# SKAP2—A Molecule at the Crossroads for Integrin Signalling and Immune Cell Migration and Function

**DOI:** 10.3390/biomedicines11102788

**Published:** 2023-10-14

**Authors:** Marijn Wilmink, Marianne Rebecca Spalinger

**Affiliations:** Department for Gastroenterology and Hepatology, University Hospital Zürich, Sternwartstrasse 14, 8091 Zürich, Switzerland; marijn.wilmink@usz.ch

**Keywords:** integrin signalling, Src-kinase associated protein 2, SKAP55-HOM, macrophage function

## Abstract

Src-kinase associated protein 2 (SKAP2) is an intracellular scaffolding protein that is broadly expressed in immune cells and is involved in various downstream signalling pathways, including, but not limited to, integrin signalling. SKAP2 has a wide range of binding partners and fine-tunes the rearrangement of the cytoskeleton, thereby regulating cell migration and immune cell function. Mutations in SKAP2 have been associated with several inflammatory disorders such as Type 1 Diabetes and Crohn’s disease. Rodent studies showed that SKAP2 deficient immune cells have diminished pathogen clearance due to impaired ROS production and/or phagocytosis. However, there is currently no in-depth understanding of the functioning of SKAP2. Nevertheless, this review summarises the existing knowledge with a focus of its role in signalling cascades involved in cell migration, tissue infiltration and immune cell function.

## 1. Introduction

The immune system encounters daily challenges from microbes in the environment, and proper location and activation of immune cells is essential for defeating invading pathogens and maintaining health without initiation of overshooting or persistent inflammation [1]. In this process, mechanisms that fine-tune immune cell recruitment into tissues and that orchestrate their activation are crucial. One molecule that recently obtained interest for its role in macrophage recruitment into tissues is Src-kinase associated protein 2 (SKAP2) [2].

SKAP2, also known as SKAP55-homologue (SKAP55-HOM), is an intracellular adaptor protein that interacts with several intracellular signalling molecules. It is expressed in almost all cell types of the body; however, there is a predominant presence in immune cells [3]. The main function of SKAP2 as an adaptor protein is to facilitate the assembly of signalling complexes by interacting with multiple proteins and coordinating their activation status. By doing so, SKAP2 regulates various cellular functions and processes, including cell migration [4].

One key role of SKAP2 is the regulation of cytoskeletal dynamics, which is crucial for immune cell motility, migration, and morphology. Of note, it interacts with signalling molecules downstream of integrins, actin-binding proteins, and other cytoskeletal components to influence changes in cell shape, and thereby influencing extravasation, adhesion, and migration into tissues and sites of inflammation. SKAP2 further participates in integrating signals from surface receptors and transmitting them to downstream signalling molecules and is involved in the formation and stabilisation of immune synapses, where it contributes to the interaction between adaptive immune cells and antigen-presenting cells [5].

The genetic locus of SKAP2 on chromosome 7p 15.2 harbours several single nucleotide polymorphisms (SNPs) with significant association with inflammatory conditions, highlighting the role of SKAP2 in inflammatory processes. rs2030136, for instance, has been linked to chronic venous disease [6] and rs7804356 has been associated with type 1 diabetes [7], while rs10486483 has been linked to Crohn’s disease, one of the main forms of inflammatory bowel disease (IBD) [8,9]. Collectively, these findings suggest that SKAP2 may play a crucial role in regulating inflammatory processes and in fine-tuning immune cell location and activation in tissues. Here we summarise the current knowledge of SKAP2, with a focus of its role in signalling cascades involved in cell migration and tissue infiltration.

## 2. Structure of SKAP2

Structurally, SKAP2 consists of several distinct domains that mediate its functions. These domains include an N-terminal dimerization (DM) domain, a Pleckstrin homology (PH) domain, and a C-terminal Src homology 3 (SH3) domain (Figure 1).

The DM domain of SKAP2 contains multiple proline-rich regions, which serve as binding sites for other signalling molecules to facilitate the recruitment and activation of downstream signalling components. While mediating interactions with binding partners and dimerization of SKAP2, the DM domain also mediates a regulatory role. Studies have demonstrated that the DM domain forms a helical hairpin consisting of α1 and α2 helices. Upon interaction with another SKAP2 protein, the DM domain of one molecule forms a homodimer with the DM domain of its counterpart, resulting in the formation of a four-helix bundle [10]. Studies examining SKAP2 with a mutant DM domain revealed impaired actin polymerisation, resulting in reduced actin ruffling, a process that involves the dynamic rearrangement of actin filaments to form membrane protrusions. These actin ruffles are important for various cellular functions, including cell migration, adhesion, and signalling. Thus, the DM domain is importantly involved in the function of SKAP2 to mediate immune cells recruitment and activation.

The centrally located PH domain is responsible for membrane localisation, protein-protein, and protein–lipid interactions. The PH domain is a highly conserved protein module found in numerous signalling proteins and acts as a binding site for specific phosphoinositides, such as phosphatidylinositol (3,4,5)-trisphosphate (PIP3) [11], a lipid second messenger formed as a product of phosphatidylinositol 3-kinase (PI3K) in response to tyrosine kinase activity. The most well-known PIP3 signalling pathway involves Akt and promotes cell proliferation, survival, and inhibition of apoptosis [12].

The interaction between the PH domain of SKAP2 and PIP3 has shown to be important for its subcellular relocation to the plasma membrane. A specific point mutation in the PH domain in SKAP2, where the Arginine at position 140 was replaced with a Methionine (R140M), disrupts the binding of PIP3 to SKAP2. As a consequence, this mutation negatively affects the relocation of SKAP2 to the plasma membrane and prevents its proper function, finally resulting in impaired actin ruffling [10]. The R140M mutation mimics the effects of PI3K inhibition, which implies that PIP3 binding to the PH domain is crucial in the initiation of actin ruffling and cell motility. Conversely, elimination of the entire PH domain gave rise to indistinguishable actin ruffles when compared to wild type (WT) SKAP2 [10]. This unforeseen observation challenged the prevailing notion that the PH domain of SKAP2 was essential for the modulation of actin dynamics.

In resting state, the DM domain and the PH domain of SKAP2 interact, which physically obstructs the binding site for PIP3 on the PH domain. Thus, the DM-PH structure barricades the binding of SKAP2 with PIP3 on the plasma membrane. However, elevated levels of PIP3 can free the DM-PH binding, and thus releasing the auto-inhibitory effect of the DM-PH domain (Figure 1) [4]. Interestingly, disrupting the DM-PH interaction with a point mutation (replacing Aspartate at position 129 with a Lysine; D129K mutation), did not only increase PIP3′s binding affinity to SKAP2 [10], but led to hyperactive actin polymerisation in macrophages [4]. Moreover, when disrupting PIP3′s binding pockets and simultaneously hindering the DM-PH interaction with a R140M/D129K double mutant, actin polymerisation appeared normal or slightly elevated when compared to WT SKAP2 [4,10]. Taken together, these findings imply that the DM-PH domains and their interaction serve as a regulatory mediator that interferes with PIP3 binding and auto-inhibits actin polymerisation in steady state.

Finally, the C-terminal, SH3 domain of SKAP2 is involved in protein–protein interactions by interacting with proline-rich motifs found in other proteins, including adhesion and degranulation-promoting adapter protein (ADAP) and the tyrosine kinase Fyn, a member of the Src family kinases (SFKs) [13,14]. The interaction between SKAP2 and its binding partners through the SH3 domain is important for the regulation of immune cell activation and for mediating the function of SKAP2 binding partners in cell adhesion dynamics. Overall, SKAP2 functions as a linker protein that connects upstream signalling molecules to downstream effectors, thereby transmitting signals and coordinating cellular responses.

## 3. Binding Partners of SKAP2

As mentioned above, important effector pathways that involve SKAP2 are integrin engagement [7], actin polymerisation, cytoskeletal rearrangement, and regulation of immune cell migration and function (Figure 2) [4,15]. In the following paragraph, we describe what is known about SKAP2′s involvement in these pathways.

### 3.1. Integrin Activation and Integrin Signalling

Cell motility relies on the dynamic rearrangement of the cytoskeleton, particularly the dynamics of actin filaments. These structural components provide the necessary framework for cellular movement. However, for cells to effectively interact with their surroundings and migrate, they require a connection to the extracellular matrix (ECM) through transmembrane cell adhesion proteins. One prominent group of adhesion molecules involved in this process are integrins. Integrins are heterodimeric proteins composed of an α and a β subunit, each with multiple isoforms [16,17]. The combination of specific α and β subunits determines the binding specificity of integrins for ECM components such as fibronectin, collagen, laminin, and vitronectin [16,17]. Since integrins span the cell membrane and are capable of activating intracellular signalling cascades, they act as a link between the ECM and the cytoskeleton [18]. Integrin binding facilitates the attachment of cells to the ECM, thereby facilitating the transmission of mechanical forces and signalling cues between the external environment and the cell.

Although integrins themselves lack enzymatic activity, they play a crucial role in cell signalling by interacting with various extracellular ligands [16]. In their inactive state, integrins are in a bent conformation, hiding the ligand-binding site, and creating a low-binding affinity to the ECM [16]. Extension of the integrin dimers exposes the ligand-binding site, inducing an active state of the integrins. Two of the primary proteins that induce this structural change are Talin-1 and Kindlin, which first have to be activated intracellularly [16,17,19]. These integrin activators engage with the cytoplasmic tail of β integrins and induce the conformational change. The activation of integrins via a series of intracellular events is known as inside-out signalling. On the other hand, binding of ligands to the extracellular domain of the activated integrins initiates the so called ‘outside-in’ signalling, which also induces conformational changes in the receptor, leading to the recruitment and activation of intracellular signalling proteins (Figure 3) [20].

The major signalling pathways associated with integrins include focal adhesion kinase (FAK) signalling, PI3K-Akt signalling, Rho GTPase signalling, and mitogen-activated protein kinase (MAPK) signalling [18,21].

Principally, activated FAK recruits and phosphorylates SFKs; however, it is also able to recruit PI3K. While FAK activation is not an absolute requirement for PI3K-Akt pathway activation, it can enhance and contribute to the signalling output of the pathway. FAK, and also SFK phosphorylation, can potentiate PI3K-Akt signalling by promoting the recruitment and activation of PI3K at FAs, leading to increased PIP3 production and subsequent Akt activation (Figure 3) [22]. Additionally, the FAK-SRC signalling pathway can also directly and indirectly influence the MAPK pathways [21,23].

The activation of these signalling cascades influences cell behaviour, regulating cytoskeletal rearrangement, cell migration, proliferation, and survival. By coordinating integrin-mediated adhesion, cytoskeletal reorganisation, and biochemical signalling, cells can dynamically adjust their shape and position in response to their environment, enabling vital processes such as tissue repair. Immune cells, in particular, heavily depend on adequate integrin signalling for their function and their homing to specific tissues.

SKAP2 interferes with numerous steps of the aforementioned signalling cascades. An important binding partner of SKAP2 within the integrin signalling cascade is Talin-1. Binding of Talin-1 to β-integrin does not only induce conformational changes, but can also bind directly to the actin cytoskeleton and thereby influence cytoskeletal rearrangement. Additionally, Talin-1 has been shown to function as a scaffolding protein [19]. Through its association with Talin-1, SKAP2 promotes clustering and activation of integrins at the cell membrane. Its interaction with Talin-1 enhances the affinity of integrins for extracellular matrix proteins, leading to the formation of stable focal adhesions (FA) (Figure 4) [18,20,24].

Ultimately, SKAP2 has earned its name by being associated with Src kinases, which are recruited to FA complexes to mediate integrin signalling. Src kinases consist of a SH2, a SH3, and a tyrosine kinase catalytic domain. The structure of the SRC kinases into an open or closed conformation regulates their activation. An open conformation activates SRC kinases, subsequently allowing their tyrosine phosphorylation, which stabilises their kinase activity [3].

### 3.2. Interaction with the WAVE Complex Regulates Cytoskeleton Rearrangement

Apart from the binding of SKAP2 with Talin-1, SKAP2 also interacts with the Wiskott–Aldrich syndrome protein Verprolin homologue (WAVE) complex, another molecule that connects to the actin filament. The WAVE regulatory complex is composed of the five subunits ABI (ABI1, ABI2, or ABI3), WAVE (WAVE1, WAVE2, or WAVE3), Nap1 (Nckap1 or Nckap1), CYFIP (CYFIP1 or CYFIP2), and HSCP300 and is part of the Wiskott–Aldrich syndrome protein (WASP) family [25].

Like SKAP2, the WAVE complex exhibits an auto-inhibitory conformation at resting state. Binding of GTPases from the Rho family destabilises this conformation and exposes the binding site for actin related protein (ARP) 2/3, which forms a complex and recruits actin monomers to form branched actin filaments (Figure 5) [25]. By interacting with the WAVE complex, SKAP2 modulates its activity and downstream functions.

One of the known effects of SKAP2 on the WAVE complex is the regulation of T cell activation. SKAP2, along with ADAP, forms a complex with SLP-76 (SH2 domain-containing leukocyte protein of 76 kDa) and promotes the recruitment and activation of the WAVE complex during T cell receptor signalling. This interaction enhances actin polymerisation and contributes to the formation of immune synapses between T cells and antigen-presenting cells [26].

Beyond its role in T cell biology, the WAVE complex and SKAP2 have also been implicated in other cellular processes. In neuronal development, SKAP2 influences WAVE complex-mediated axon guidance and dendritic spine morphogenesis, thus contributing to the establishment of neuronal connectivity [25,27]. SKAP2 has further been implicated in platelet activation and thrombus formation, playing a role in haemostasis and thrombosis [28]. These findings highlight the versatility of the WAVE complex and SKAP2 in different cell types and their contribution to diverse physiological processes.

Furthermore, mutations or dysregulation of SKAP2 and WASP have been linked to autoimmune diseases such as diabetes [7]. Understanding the precise mechanisms of the WAVE complex and SKAP2 regulation and their roles in disease pathology may offer potential therapeutic targets for modulating immune responses, neuronal development, and platelet function.

### 3.3. SKAP2 and Cell Migration

Given the interaction with integrins, FA turnover, and the WAVE complex, SKAP2 influences actin polymerisation and subsequently cell migration [27]. Via activation of integrins, SKAP2 promotes the binding of cells to the ECM. This activation allows cells to generate traction forces and facilitates their movement during migration. One way for cells to allow for this movement is by focal adhesion turnover. In this process, SKAP2 facilitates recruitment and activation of SFKs, which phosphorylate FA proteins, leading to their turnover and ultimately allowing cells to detach and re-attach, thus creating movement [29]. In addition, through interactions with actin-binding proteins, such as Talin-1 and Kindlin [15], SKAP2 regulates the formation of lamellipodia and filopodia. These are cellular protrusions involved in cell migration [30]. SKAP2 also interacts with the Arp2/3 complex, which promotes actin polymerisation and contributes to the generation of branched actin networks that are necessary for cell motility (Figure 5) [31]. Lastly, due to its association with SFKs, SKAP2 can promote downstream signalling pathways, such as the PI3K/Akt pathway, which regulates cell migration. SKAP2 also interacts with other signalling molecules, including SLP-76, which are involved in the activation of small GTPases and actin remodelling [13,32].

## 4. The Role of SKAP2 in Immune Cells

Utilising mouse SKAP2-knockout (KO) models have helped to understand the role of SKAP2 in migration and function of immune cells [32,33]. In the following paragraph, we summarise how SKAP2 modulates recruitment and effector functions of neutrophils, macrophages, and lymphocytes.

### 4.1. Neutrophils

It has recently been demonstrated that SKAP2 KO neutrophils are impaired in the production of reactive oxygen species (ROS) [32], which is an important effector function of neutrophils to defeat invasive microbes. ROS are produced as a by-product of the NADPH oxidase complex and can act as an antimicrobial agent by directly damaging DNA molecules. Receptor-mediated ROS production occurs when integrins, G protein coupled receptors (GPCRs), or Fc receptors are stimulated and activate the NADPH oxidase complex [34]. Due to its interaction with integrins and SFK, SKAP2 affects ROS production, which—especially in neutrophils—promotes the ability to clear invasive bacteria.

Even though SKAP2 KO mice had normal levels of circulating neutrophils and no defects in basal ROS production when compared to WT mice, SKAP2 was required for integrin-, Fc receptor- and GPCR-mediated ROS production [32]. Notably, whilst GPCR-mediated ROS production of neutrophils was impaired due to dephosphorylation of SKAP2, release of neutrophilic granules was not affected by loss of SKAP2 [32].

Similarly, SKAP2 KO mice showed normal degranulation capacity and numbers of neutrophils and monocytes present at infection sites. However, SKAP2 KO neutrophils showed a significant reduction in ROS production [33]. In this study, it was shown that SKAP2 loss resulted in impaired activation of SFK and Syk, leading to a reduction in phosphorylation of Pyk2, ultimately mediating impaired ROS production. Additionally, integrin-mediated ERK and Akt phosphorylation was decreased in SKAP2 KO neutrophils, which also contributed to reduced ROS production in these cells [13,14].

### 4.2. Macrophages

Macrophages are innate immune cells that recognise invading pathogens and efficiently initiate protective immune responses if they are not able to directly eliminate the pathogens. Furthermore, they are required for normal wound healing and removal of dead cells and debris in tissues in an immune silent manner. To perform these tasks, macrophages relay on efficient phagocytosis, a process heavily dependent on normal actin filament remodelling. Furthermore, macrophages show a diverse range of activation states, ranging from pro-inflammatory (called M1) cells that produce inflammatory cytokines and promote immune activation to immune suppressive/wound-healing-promoting (called M2) cells that promote tissue repair and remodelling. SKAP2 interacts with several pathways involved in phagocytosis and differentiation of macrophages.

#### 4.2.1. Effects on Phagocytosis in Macrophages

To initiate immune responses, macrophages need to recognise and phagocytose intruding pathogens. Engulfing pathogens requires remodelling of the actin cytoskeleton, initiated by the formation of protein complexes consisting of Fyb/SLAP, SLP-76, ADAP, VASP, profiling, and WASP [35,36]. Interference with these protein complexes leads to inhibition of phagocytosis, and activation of SKAP2 seems to be an important mediator in this process. The effector protein YopH, produced by the pathogenic species of *Yersinia*, has shown to be capable of reducing SKAP2 activation, thereby inhibiting phagocytosis, as well as hindering adhesion-regulated signalling transduction pathways in macrophages [37].

Another way for SKAP2 to influence phagocytosis in macrophages is by interacting with the inhibitory surface receptor signal regulatory protein α (SIRPα). SIRPα, expressed on myeloid cells, binds to CD47, a molecule also known as the ‘do-not eat me’ receptor as it suppresses phagocytosis by macrophages and other phagocytic cells. When SIRPα is engaged by its ligand CD47, its immunoreceptor tyrosine-based inhibition motif (ITIM) is phosphorylated and serves as a docking site for SHP1 and SHP2 [38]. SHP1 and SHP2 are two closely related enzymes known as protein tyrosine phosphatases (PTPs). The binding of SHP1 or SHP2 to the phosphorylated ITIM of SIRPα leads to the activation of the phosphatase activity of these enzymes. Once activated, SHP1 and SHP2 remove phosphate groups from specific tyrosine residues on target proteins, thereby modulating downstream signalling events (Figure 6). Thus, when SIRPα is linked with CD47 on neighbouring cells, SHP1 and SHP2 are activated, resulting in the suppression of immune cell activation and phagocytosis [38,39].

SIRPα KO mice show impaired wound healing, associated with defective alternative macrophage (M2 macrophage) induction in experimental colitis [40]. Mechanistically, binding of CD47 to SIRPα promoted phosphorylation of SIRPα’s ITIM domain and caused recruitment of SHP-2 to SIRPα. This interaction prevented SHP-2 from binding and dephosphorylating other targets such as the IL-4 and IL-13 receptors. Loss of SIRPα resulted in inhibiting the IL-4R and IL-13R, leading to a dampening of IL-4 and IL-13 signalling, which normally promotes the activation of alternative macrophages and thereby stimulates wound healing [40].

In the context of the interaction between SIRPα and SHP1/SHP2, SKAP2 can associate with SIRPα through its SH3 domain. Once SKAP2 binds to SIRPα, it helps facilitate the interaction between SIRPα and SHP1/2, enhancing dephosphorylation of their target proteins and promoting the proper regulation of downstream signalling pathways. Thus, SKAP2 acts as a bridging molecule, connecting SIRPα and SHP1/SHP2 (Figure 6). Apart from forming a complex with SIRPα and SHP1/2, SKAP2 also physically associates with MyD88, TIRAP, and TRAM, adaptors of the Toll Like Receptor 4 (TLR4) [41]. SKAP2 is capable of bridging and mediating the recruitment of the SIRPα and SHP1/2 complex to TLR4, which leads to the attenuation of an inflammatory response in an experimental colitis mouse model [41].

Disruption of the interaction between SKAP2 and SIRPa has important implications for diseases such as cancer, autoimmune disorders, and inflammation. It has already been shown that recruitment of SHP1/2 mediated the inhibition of phagocytosis in macrophages [42] and that loss of SKAP2 decreased the activity of SIRPα and reduced actin ruffling in macrophages [4].

#### 4.2.2. SKAP2 and the Differentiation of Macrophages

It has also been implied that SIRPα and SKAP2 are required for macrophage differentiation into M2 macrophages. In SIRPα deficient mice, a reduction in alternative activated (M2) macrophages was found to be associated with impaired wound healing in experimental colitis [40]. On the other hand, two separate studies have found that SKAP2 deficient mice exhibited accelerated atherogenesis due to reduced M2 macrophage polarisation and enhanced levels of classically activated (M1) macrophages [43,44]. Furthermore, studies evaluating SKAP2 deficient macrophages show that bone-marrow-derived macrophages (BMDMs) from SKAP2 KO mice exhibit impaired integrin signalling, migration, and actin polymerisation [16]. Nevertheless, there are some controversial studies regarding its effect on inflammation. Experimental autoimmune encephalomyelitis (EAE) was found to be less severe in SKAP2 KO mice compared to WT mice. So far, this is the only study that revealed a decrease in inflammation due to SKAP2 KO, while the majority of in vivo studies using SKAP2 KO mice have shown that SKAP2 KO mice are more susceptible to inflammation [41].

### 4.3. Lymphocytes

While macrophages play a key role in the innate immune response, T cells orchestrate a smooth functioning of the adaptive immune response. For full activation, T cells require interaction with an antigen-presenting cell (APC), such as macrophages or dendritic cells. Between these immune cells, an immunological synapse is formed to facilitate the signalling required for sufficient T cell activation. The formation of the immunological synapse also relies on remodelling of the cytoskeleton and requires the activation of several SKAP2 binding proteins including the WASP protein [45], SLP-76, and PIP3 [46].

Remarkably, T cells are one of the few cell types that expresses a SKAP2 homologue, SKAP1. SKAP1 and SKAP2 share a great structural similarity, yet they can be distinguished by their tyrosine-phosphorylated state in resting human T cells. Whereas SKAP2 needs external stimuli to become phosphorylated, a recent study revealed that SKAP1 is constitutively phosphorylated [14]. The binding partner shared between SKAP1 and SKAP2 is ADAP. In T cells, ADAP is known to stabilise SKAP1 and prevent its degradation [47]. Phosphorylation of SKAP1 and ADAP activates a signalling cascade that influences actin polymerisation and cell motility. Interestingly, some studies imply that SKAP1 can take over the function of SKAP2, as SKAP2 KO mice do not display any overt T cell defects [48]. On the other hand, is has been found that only SKAP1 is capable of regulating lymphocyte function-associated antigen (LFA)-1 clustering on T cells and that this cannot be replaced by SKAP2 [49].

In mature B cells, neither SKAP1 nor ADAP are expressed and SKAP2 seems to play a more important role in B cell function than in T cells. SKAP2 KO B cells exhibit reduced B cell receptor-mediating proliferation and defective LFA-1 mediated clustering and adhesion [48]. However, further details regarding the involvement of SKAP2 in B cell receptor function have not been studied to date. Excitingly, the missing ADAP expression in B cells implies that there might be other interaction partners that stabilise SKAP2, and the effector molecule Rap1 interacting adaptor molecule (RIAM) has been proposed to fulfil this role [50].

## 5. Relevance for Disease Development

### 5.1. Cancer

Most of the aforementioned binding partners of SKAP2 have already been researched in the context of cancer. Interestingly, expression of SKAP2 has been associated with a poorer prognosis in lung cancer [51] and aberrant activity of SKAP2 has been observed in various cancer cell types. SKAP2 has been found to be tyrosine phosphorylated in macrophages when co-cultured with cancer cells [52]. In early stages, cancer cells are phagocytosed by macrophages, which prevents tumour formation. However, cancer cells have developed many ways to evade this ultimatum. Tumours have been reported to release growth factors such as M-CSF, and cytokines including IL4 and IL10, which initiate differentiation of macrophages into tumour-associated macrophages (TAMs) [52]. The presence of TAMs has been associated with a poor prognosis in cancer [53]. The formation of TAMs is promoted when the macrophages form actin rich structures called podosomes. Podosomes have a primary purpose to adhere to and degrade the ECM by connecting the cytoskeleton to cell surface integrins forming outward protrusions. Distinctly present in podosomes is the aforementioned WASP protein [52]. Studies revealed that SKAP2 is required for podosome formation by physically interacting with WASP. This has been confirmed when SKAP2KO macrophages hardly formed podosomes.

Lastly, the PI3K/Akt pathway, which has previously been linked to SKAP2, has been found to affect tumour cell invasiveness. Results suggest that Akt activation increases invasiveness of tumours by increasing cellular pressure, which ultimately stimulates adhesion [54]. The effect of SKAP2 on this phenomena has not yet been clarified, but inhibition of FAK, Src, and PI3K (which are all known to stimulate Akt activation) have all lead to inhibiting pressure-induced adhesion of various cancer cells [22]. These results imply that SKAP2 positively regulates tumour invasion, whilst conversely negatively influencing inflammatory disorders.

### 5.2. Inflammatory Disorders

Genetic SKAP2 variants are associated with several inflammatory disorders, including type I diabetes [7], chronic venous disease [6], and Crohn’s disease [8,9]. It has been suggested that SKAP2 is involved in regulating β-cell apoptosis in the pancreas and mediate glycaemic control in newly diagnosed diabetes patients [55], and a study that addressed the role of SKAP2 in colitis-associated cancer showed that SKAP2 loss promotes inflammatory processes downstream of TLR4 via interaction with SHP1 and SHP2 [41]. However, besides these two studies, very little is known about the mechanisms by which SKAP2 affects the development of inflammatory disorders.

## 6. Concluding Remarks

Via integration of several signalling cascades within the cell, SKAP2 promotes cell motility, tissue migration, and crucial functions of immune cells, such as ROS production and phagocytosis. In this way, SKAP2 is involved in various cell functions, and while its dysfunction does not directly cause severe disease, it is required for fine-tuning the immune system’s reaction to pathogens. The mechanisms by which SKAP2 affects inflammatory diseases has not been studied in detail. Additional studies to reveal the mechanisms by which SKAP2 affects tissue inflammation will be of great interest to further understand the various roles of SKAP2 and to better understand the fine-tuning mechanisms in the body’s immune system that prevent the emergence of inflammatory disorders.

## Figures and Tables

**Figure 1 biomedicines-11-02788-f001:**
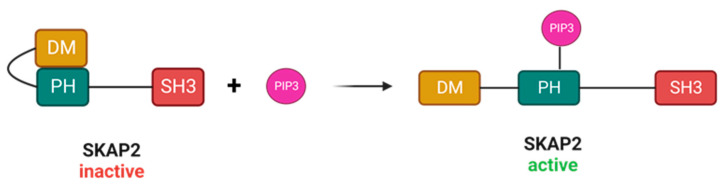
SKAP2 consists of a DM domain, a PH domain, and a SH3 domain. In resting state, SKAP2 has an auto-inhibitory state where the DM domain binds to the PH domain. PIP3 can interact with the PH domain, thereby disrupting the DM-PH structure and releasing SKAP2′s inhibitory state. SKAP2: Src-kinase associated protein 2, DM: dimerization domain, PH: Pleckstrin homology, SH3: Src homology 3, PIP3: phosphatidylinositol (3,4,5)-trisphosphate.

**Figure 2 biomedicines-11-02788-f002:**
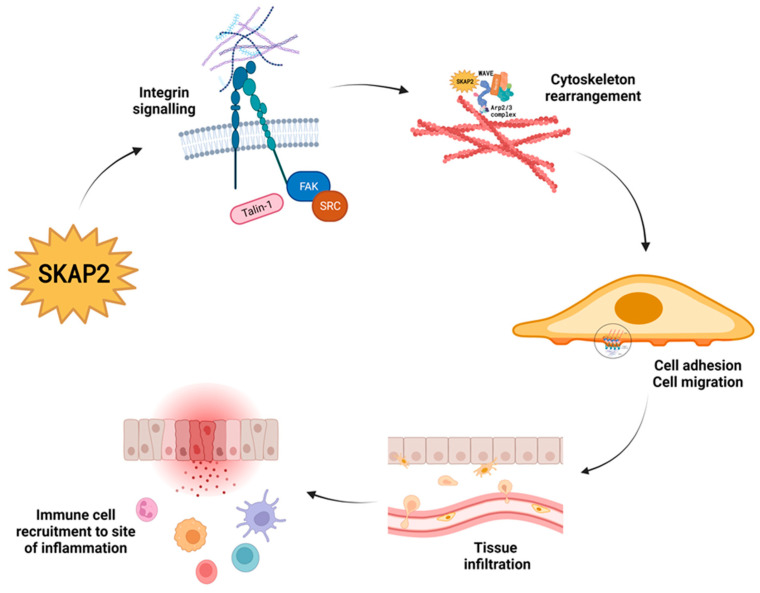
Src-kinase associated protein 2 (SKAP2) is an intracellular adaptor protein that mediates cellular responses such as integrin signalling, cytoskeletal rearrangement, cell adhesion and migration, and ultimately, regulation of immune cell recruitment and function.

**Figure 3 biomedicines-11-02788-f003:**
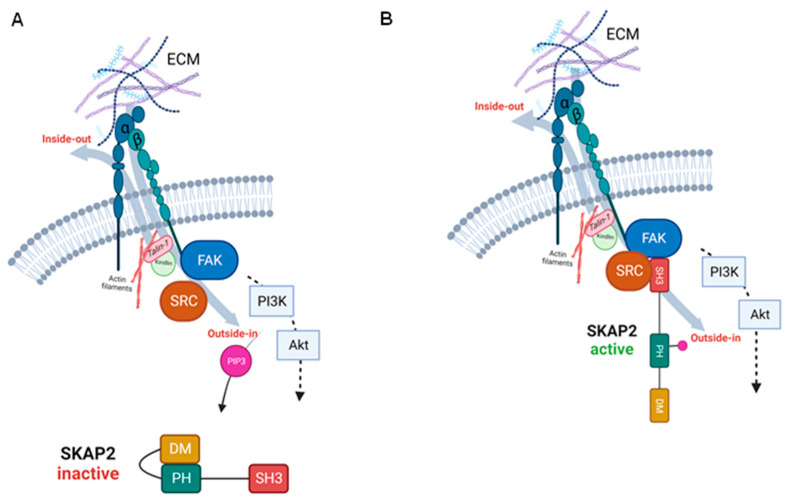
(**A**) Integrins can be activated extracellularly, via outside-in signalling or via a series of intracellular events called inside-out signalling. Upon integrin activation, FAK and SRC-kinases are recruited and phosphorylated, which stimulates the production of PIP3 via activation of the PI3K-Akt pathway. PIP3 can release the auto-inhibitory state of SKAP2. (**B**) Activated SKAP2 binds to Src-kinases and enhances downstream signalling pathways. FAK: focal adhesion kinase, SRC: Src-kinase, PIP3: phosphatidylinositol (3,4,5)-trisphosphate, PI3K: phosphatidylinositol 3-kinase, Akt: protein kinase B, SKAP2: Src-kinase associated protein 2.

**Figure 4 biomedicines-11-02788-f004:**
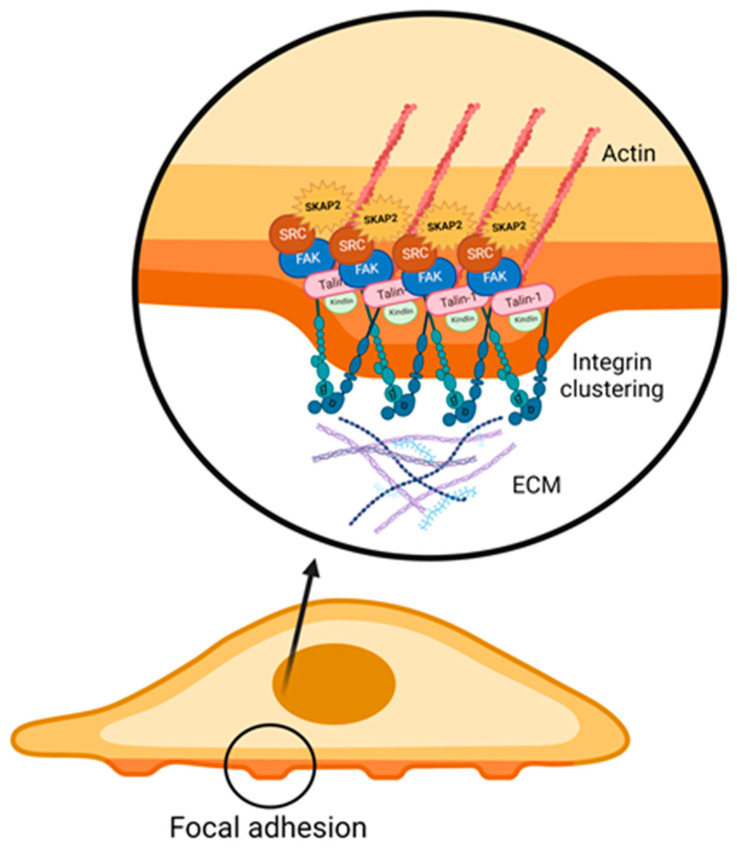
SKAP2 enhances clustering of integrins during the formation of focal adhesion, necessary for cell motility and cell adhesion, by promoting the interaction with Talin-1 and the cytoskeleton (actin). SKAP2: Src-kinase associated protein 2.

**Figure 5 biomedicines-11-02788-f005:**
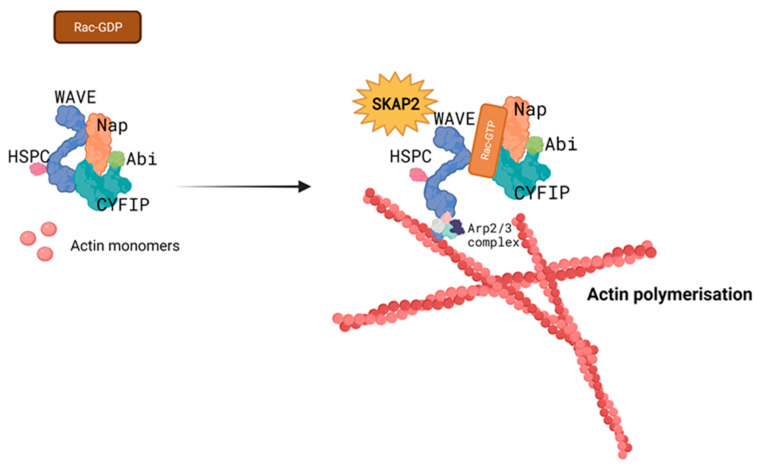
The WAVE regulatory complex consists of the five subunits ABI (ABI1, ABI2, or ABI3), WAVE (WAVE1, WAVE2, or WAVE3), Nap1 (Nckap1 or Nckap1), CYFIP (CYFIP1 or CYFIP2), and HSCP300, and is part of the WASP family. Interaction with GTPases releases the complex’s auto-inhibitory state and reveals the binding site for the Arp2/3 complex on the WAVE subunit. When activated, Arp2/3 polymerises actin monomers into actin branches, enhancing the cytoskeleton network. SKAP2 is known to interact with the WAVE complex to stabilise its active state and thereby promote cell movement. WAVE: the Wiskott–Aldrich syndrome protein Verprolin homologue, WASP: Wiskott–Aldrich syndrome protein, Arp2/3: actin related protein 2/3, SKAP2: Src-kinase associated protein 2.

**Figure 6 biomedicines-11-02788-f006:**
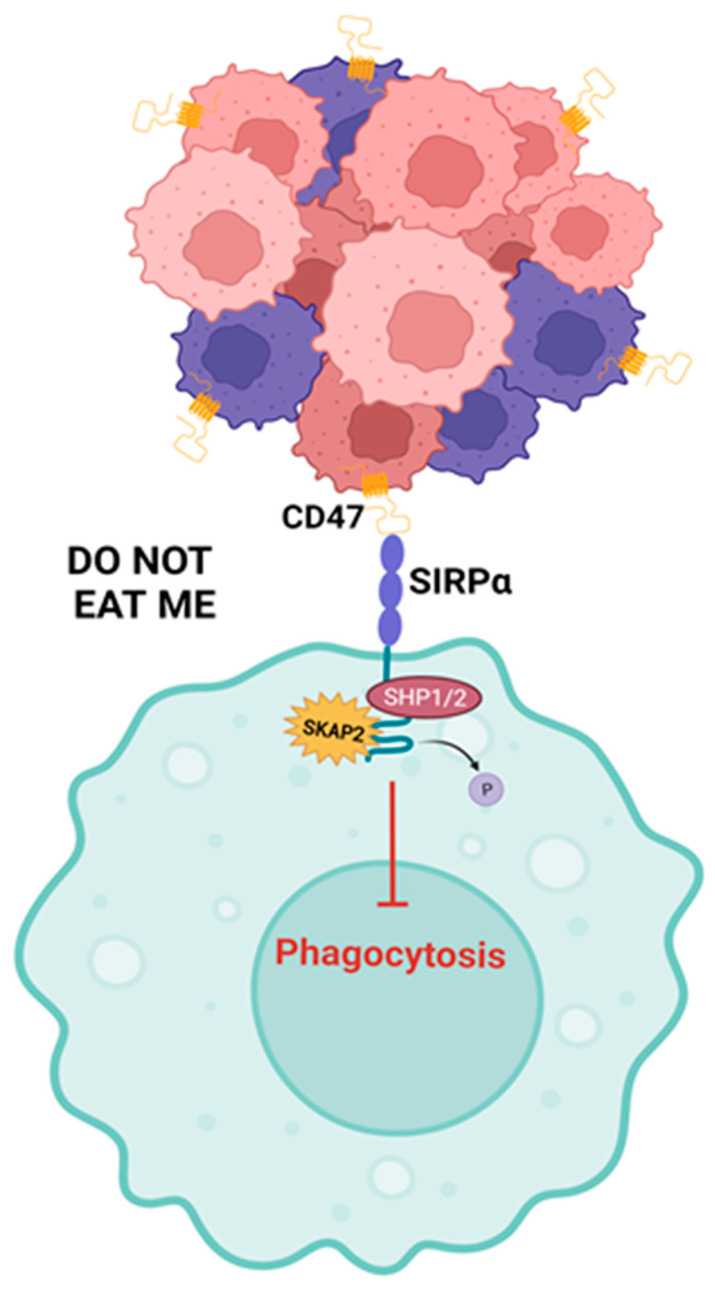
The interaction of CD47 with SIRPα forms the ‘do-not eat me’ signal and suppresses phagocytosis of the CD47-expressing cells. Recruitment and activation of SHP1/SHP2 dephosphorylates SIRPα and inhibits the initiation of phagocytosis. SKAP2 associates with SIRPα and augments the interaction of SHP1/SHP2 with SIRPα, boosting the inhibition of phagocytosis. SIRPα: signal regulatory protein α, SHP1/2: Src homology 2-containing protein tyrosine phosphatase-1/2.

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
