# Peer review of "SKAP2—A Molecule at the Crossroads for Integrin Signalling and Immune Cell Migration and Function"

_biomedicines, 2023, doi:10.3390/biomedicines11102788_

Round 1
Reviewer 1 Report
Thank you for the opportunity to review this valuable manuscript. The authors have provided an intriguing insight into the role of SKAP-2a molecule in "Outside-in" and "Inside-out" signaling of integrins, which is of great interest to experts in the field of integrins.
Furthermore, this manuscript serves as an informative introduction to the fundamental aspects of SKAP-2a for a broader readership.
I have the following concerns:
Major comment.
#1 Additional figures are required.
It would be beneficial for the authors to illustrate how the SKAP-2a molecule is involved in the activation mechanism of integrins. Specifically, a figure summarizing how SKAP-2a is implicated in both outside-in and inside-out signaling should be included.
Minor comments.
#1 Inconsistency in font usage.
There is inconsistency in the typeface used throughout the manuscript. For instance, the font used in lines 254 to 265 is different from that in lines 266 to 281, and similar instances are scattered throughout the text. Please standardize the font usage.
#2 Correct highlighted references.
The reference numbers in lines 233 and 295 are formatted as highlighted text. These should be corrected.
N/A
Author Response
First of all, we wish to thank all the reviewers for their helpful comments and suggestions on how to improve our review. We have now addressed the suggestions and amended our manuscript accordingly. Answers to the specific comments of the reviewers are given below. Changes to the original version of our manuscript are highlighted using the “Track Changes” function in Microsoft Word.
Reviewer 1:
Major comment.
#1 Additional figures are required.
It would be beneficial for the authors to illustrate how the SKAP-2a molecule is involved in the activation mechanism of integrins. Specifically, a figure summarizing how SKAP-2a is implicated in both outside-in and inside-out signaling should be included.
Authors response: We thank the reviewer for this very helpful suggestion and have now generated new figures, including a figure on SKAP2’s role in outside in and inside out integrin signaling.
Minor comments.
#1 Inconsistency in font usage.
There is inconsistency in the typeface used throughout the manuscript. For instance, the font used in lines 254 to 265 is different from that in lines 266 to 281, and similar instances are scattered throughout the text. Please standardize the font usage.
Authors’ response: We have now made sure that the font is consistent in the whole text.
#2 Correct highlighted references.
The reference numbers in lines 233 and 295 are formatted as highlighted text. These should be corrected.
Authors’ response: The highlight in the mentioned references has been removed.
Reviewer 2 Report
This review article contains summarized descriptions on the molecular and structural characteristics of SKAP2 and its roles in regulating integrin signaling and immune cells’ dynamic functions and further some inflammatory disorders. Current form of the manuscript remains a worth revisiting outlined knowledge on SKAP2 biology;including additional figure and corresponding descriptions would scientifically better convey authors’ messages on functionally diverse aspects of SKAP2. Figure 1 shows multifaceted roles of SKAP2 in a variety of physiological and pathological conditions. It can be fine with the current Figure if it is enough to comprehend the SKAP2 engaged in various processes. But, as a strong suggestion, it is recommended to include a new figure that could make it possess any uniqueness as a review article. For example, this new illustration can contain intracellular integrin signaling pathways that include SKAP2 as a key adaptor along with other multiple adaptors under both conditions to eliciting physiological (homeostatic, regulatory, or anti-inflammatory) and pathological (pro-inflammatory) pathways, if available. Also, some paragraphs, including the paragraph starting with line 31, the paragraph starting with line 145, the paragraph starting with line 235, appear to need more references.
Author Response
Reviewer 2:
This review article contains summarized descriptions on the molecular and structural characteristics of SKAP2 and its roles in regulating integrin signaling and immune cells’ dynamic functions and further some inflammatory disorders. Current form of the manuscript remains a worth revisiting outlined knowledge on SKAP2 biology;including additional figure and corresponding descriptions would scientifically better convey authors’ messages on functionally diverse aspects of SKAP2. Figure 1 shows multifaceted roles of SKAP2 in a variety of physiological and pathological conditions. It can be fine with the current Figure if it is enough to comprehend the SKAP2 engaged in various processes. But, as a strong suggestion, it is recommended to include a new figure that could make it possess any uniqueness as a review article. For example, this new illustration can contain intracellular integrin signaling pathways that include SKAP2 as a key adaptor along with other multiple adaptors under both conditions to eliciting physiological (homeostatic, regulatory, or anti-inflammatory) and pathological (pro-inflammatory) pathways, if available. Also, some paragraphs, including the paragraph starting with line 31, the paragraph starting with line 145, the paragraph starting with line 235, appear to need more references.
Authors’ response: As suggested, we have now added additional figures on SKAP2’s role in different intracellular signaling pathways to make grasping the multifaceted role of SKAP2 easier.
Additional references have been added to the mentioned paragraphs (first paragraph in the introduction, first paragraph in section 3.1, first paragraph in section 4).
Round 2
Reviewer 1 Report
I am pleased to review this manuscript again. The authors have addressed a major concern I raised by creating a new figure to clarify the activation of integrins through the two modes, 'outside-in signaling' and 'inside-out signaling'. As a result, I believe the state of the manuscript has improved. I think it will be an intriguing read for many readers.
N/A